# Remarkable Separation of Carbofuran Pesticide from Aqueous Solution Using Free Metal Ion Variation on Aluminum-Based Metal-Organic Framework

**Marwa Nabil** [1], **Fatma M. Elantabli** [1] , **Samir M. El-Medani** [1] **and Reda M. Abdelhameed** [2,*]

1 Chemistry Department, El-Fayoum University, El-Fayoum 63514, Egypt
2 Department of Applied Organic Chemistry, Chemical Industries Research Institute, National Research Centre, 33 EL Buhouth St., Dokki, Giza 12622, Egypt
\* Correspondence: rm.ahmed@nrc.sci.eg; Tel.: +20-112-553-2445

**Abstract:** The alarming increase in pesticide residues poses a major threat to aquatic and natural habitats. Therefore, it has become essential to design extremely operationally and economically advantageous systems for the removal of carbofuran pesticides from wastewater. Here, an aluminum-based metal-organic framework (MOF), MIL-53-NH$_2$, and its modified forms, MIL-53-NH-ph, MIL-53-NH-ph-Fe, MIL-53-NH-ph-Zn, and MIL-53-NH-ph-Cu, have been successfully synthesized. Full characterization using IR, $^1$HNMR, XRD, and SEM was carried out. The prepared MOFs have been utilized as effective adsorbents for carbofuran in aqueous solutions. The various factors affecting the adsorption process (pH, time, and adsorbate concentration) were also investigated. Spectroscopic approaches were used to investigate the adsorption mechanisms. A mixture of π-π stacking contact, coordination bonding, and hydrogen bond formation can be connected to the current process. The adsorption of carbofuran from aqueous solutions was best described by pseudo-second-order kinetics and Langmuir equilibrium isotherm models. MIL-53-NH$_2$, MIL-53-NH-Ph, MIL-53-NH-Ph-Fe, MIL-53-NH-Ph-Zn, and MIL-53-NH-Ph-Cu demonstrated adsorption capacities of 367.8, 462.1, 662.94, 717.6, and 978.6 mg g$^{-1}$, respectively.

**Keywords:** free metal center; aluminum-organic frameworks; carbofuran; adsorption; wastewater

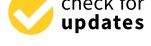



## 1. Introduction

Worldwide, huge quantities of pesticides are used to control plant diseases. They are used to kill or prevent pests from harming plants. Pesticide use causes a slew of issues for both the environment and humans. As a consequence they contaminate water and soil and cause many diseases in birds, plants, animals, and humans, including cancer, diabetes, reproductive, respiratory, and neurological disorders [1]. Carbofuran, also known as 2,3-dihydro-2,2-dimethyl-7-benzofuranol methyl carbamate, is a widely used chemical in agriculture. The toxicity of carbofuran to mammalian systems has been demonstrated and it causes concern [2]. The metabolic pathway of carbofuran in organs such as the brain, liver, muscles, and heart was studied to identify its toxicity for both humans and animals [3]. Water contaminated with substantial levels of carbofuran makes drinking water with these residues extremely dangerous [4].

Adsorption and bioremediation are ideal procedures for removing carbofuran from wastewater because they are ecologically benign, cost-effective, and produce fewer harmful by-products. Metal-organic frameworks (MOFs) have been employed to improve the recovery of pesticides from wastewater by adsorption. MOFs are porous three-dimensional frameworks with open channel systems made up of a metal ion core linked to an organic linker made up of organic molecules [5]. Nowadays, improving the physical and chemical stability of MOFs is necessary to increase their use in water treatment [6]. As a result, finding stable MOFs has become increasingly desirable. MIL-53(Al) is a new adsorbent

that has an octahedral $AlO_4(OH)_2$ node and phthalate groups as a linker. MIL-53(Al) has a three-dimensional network structure with a one-dimensional open channel [7]. Due to the huge surface area of this MOF, redesigned pores, and free active sites, pesticides have been successfully removed from wastewater using this MOF [8].

Ethion pesticide was removed from wastewater using copper-based MOFs (Cu-BTC) [9] and its adsorption capacity was found to be 0.317 mol $g^{-1}$. However, when ZIF-8 and ZIF-67 were used, the adsorption capabilities were increased by 2.29 and 1.73 times, respectively [10]. The MOFs were applied in powder form and suffer from reusability issues, so, in an attempt to facilitate the adsorption process, Cu-BTC was integrated into cotton fabric, and, thereby, demonstrated significant ethion pesticide adsorption capacity and reusability (Table 1). In addition, Cu-BTC was added into cellulose acetate, which demonstrated improved dimethoate insecticide adsorption capability [11].

**Table 1.** Adsorption of pesticides from wastewater with MOFs.

| Pesticide | MOFs | Surface Area ($m^2$ $g^{-1}$) | pH | Adsorption Capacity (mol $g^{-1}$) | Reference |
|---|---|---|---|---|---|
| Diazinon | MIL-101 (Cr) | 2600 | 7 | 0.855 | [12] |
| Ethion | Cu-BTC | 980 | 7 | 0.317 | [9] |
| Prothiofos | ZIF-8 (Zn) | 1800 | 7 | 1.07 | [11] |
| Ethion | ZIF-8 (Zn) | 1800 | 7 | 0.726 | [10] |
| Prothiofos | ZIF-67 (Co) | 1250 | 7 | 0.761 | [10] |
| Ethion | ZIF-67 (Co) | 1250 | 7 | 0.549 | [10] |
| Glyphosate | UIO-67 (Zr) | 2172 | 4 | 3.18 | [13] |
| Dimethoate | Al-MOF | 1260 | 7 | 2.24 | [14] |
| Ethion | Cu-BTC@cotton | 750 | 7 | 0.474 | [15] |
| Dimethoate | Cu-BTC@ cellulose acetate | 965.8 | 7 | 1.694 | [11] |

Prothiofos insecticide was removed from wastewater with ZIF-8 (Zn) and ZIF-67 (Co), and the metal node in the network was shown to play an important role in the removal process [10]. MIL-53 MOFs with various amino ratios have recently been employed to extract dimethoate from wastewater. When the fraction of the amino group was raised, its maximum adsorption capacity increased, demonstrating that the amino group in the Al-MOF considerably enhanced the dimethoate adsorption ability [14].

In this study, we present a water-stable MOF MIL-53-$NH_2$ and modified forms (MIL-53-NH-ph, MIL-53-NH-ph-Fe, MIL-53-NH-ph-Zn, and MIL-53-NH-ph-Cu), which were synthesized and characterized to be used as effective sorbents for carbofuran. Furthermore, the effects of adsorbate concentration, time, and pH on the adsorption process were investigated, and the adsorption mechanism was evaluated by various spectroscopic techniques.

## 2. Experimental Section

### 2.1. Materials

The aluminum chloride ($AlCl_3.6H_2O$) (99.9%, Aldrich), 2-aminoterephthalic acid (99%, Aldrich), sodium hydroxide (NaOH), phthalic anhydride (99.9%, Aldrich), dioxane (99%, Aldrich), ferrous nitrate ($Fe(NO_3)_2$) (99.9%, Aldrich), copper nitrate ($Cu(NO_3)_2$) (99.9%, Aldrich), zinc nitrate ($Zn(NO_3)_2$) (99.9%, Aldrich), and ethanol ((99.9%, Aldrich), used in this study were all analytical grade and used without further purification.

### 2.2. Synthesis of MIL-53-$NH_2$

$AlCl_3.6H_2O$ (0.66 g, 2.7 mmol), 2-aminoterephthalic acid (1.35 g, 7.5 mmol), and NaOH (0.6 g, 15 mmol) were dissolved separately in 10 mL of distilled water at room temperature. These solutions were then mixed, and the resulting slurry was sealed and heated in a hydrothermal reactor at 110 °C for 24 h. The resulting solution was then centrifuged and rinsed ten times with 10 mL of distilled water. The final product was dried at 80 °C for 4 h. The percentage yield of the synthesized materials was 45%.

### 2.3. Post-Synthetic Modification of MIL-53-NH$_2$

2.3.1. Synthesis of MIL-53-NH-ph

MIL-53-NH-ph was produced by dissolving MIL-53-NH$_2$ (1.16 g, 5.2 mmol) and phthalic anhydride (3.015 g, 20.3 mmol) in 10 mL of dioxane and heating for 24 h at 150 °C. The final product was centrifuged and washed ten times with distilled water then dried at 100 °C for 24 h. The percentage of conversion was 54%.

2.3.2. Synthesis of MIL-53-NH-ph-M (M = Fe, Cu, Zn)

MIL-53-NH-ph-M was prepared by reacting MIL-53-NH-ph (1 g, 3.3 mmol) and Fe (NO$_3$)$_2$ (4.4 g, 24.46 mmol), Cu(NO$_3$)$_2$ (2.4 g, 12.8 mmol), and Zn(NO$_3$)$_2$ (2.9 g, 12.31 mmol) in 10 mL of ethanol under stirring for 72 h at room temperature. The solution was centrifuged and rinsed ten times with distilled water. The final product was dried at 80 °C for 24 h.

### 2.4. Characterization of the Synthesized Compounds

The morphology of the MIL-53-MOFs was investigated using high-resolution scanning electron microscopy (HRSEM Quanta FEG 250 with field emission gun, FEI—Eindhoven, The Netherlands). Elemental analysis was carried out using an ener-gy-dispersive x-ray spectroscopy (EDX) device (EDAX AME-TEK Analyzer, Mawah, NJ, USA) coupled to the SEM microscope. An X'Pert PRO PAN diffractometer (Cu-Ka X radiation at 40 kV, 50 mA, wavelength = 0.15406 cm−1) was used to detect powder x-ray diffraction (PXRD) of MIL-53-NH2, MIL-53-NH-ph, MIL-53-NH-ph-Fe, MIL-53-NH-ph-Zn, and MIL-53-NH-ph-Cu. The Fourier transforms infrared (FTIR) spectra of the synthesized compounds were obtained by a JASCO FTIR-4700 spero-photometer. The 1H NMR spectrum of MIL-53-NH-ph was investigated using a Bruker AVANCE 400 MHZ spectrometer (Bruker, Billerica, MA, USA) with dimethyl sulfoxide as the solvent (DMSO-d6). The surface area and pore size of the synthesized compounds were measured at 77 K using a volumetric adsorption analyzer (ASAP 2420, Micrometrics, Norcross, GA, USA). CNH contents were determined by a LECO CHNS-932 element analyzer. The metal content of the modified MOFs samples was analyzed using an atomic absorption spectrophotometer (Perkin-Elmer Analyst 200 AAS, Waltham, MA, USA).

### 2.5. Adsorption Study

Adsorption experiments of carbofuran on the synthesized compounds were carried out at 303 K with shaking at 400 rpm. The experiments were carried out by adding adsorbent (30 mg) to different concentrations of carbofuran solution (5–30 mg) at pH = 7. The remaining concentration of the carbofuran solution was detected using ultraviolet-visible (UV–VIS) spectrophotometry (JASCO) at λ = 286.5 nm, and calculated from a calibration curve. Several factors, such as carbofuran initial concentration and equilibrium time, were studied to determine the perfect conditions for carbofuran adsorption on MIL-53-NH$_2$, MIL-53-NH-ph, MIL-53-NH-ph-Fe, MIL-53-NH-ph-Zn, and MIL-53-NH-ph-Cu. Throughout the experiment, the test tubes were removed periodically at specific time intervals (15 min–120 min) and centrifuged to separate the adsorbent. The effect of the pH of the adsorbent (MOF and its modified forms) solution was studied at pH values of 2, 7, and 9. The solution pH was adjusted using solutions of HCl (0.1 M) and NaOH (0.1 M). The pH measurements were made using a HANNA instruments digital pH meter (HI 8417 Microprocessor Bench pH/mV/°C meter). It was found that pH 7 is the optimum pH for using the prepared MOFs in the adsorption of carbofuran solution. Three replicates were conducted, and the mean value was used. The adsorbed amount ($Q_t$, mg g$^{-1}$) of the studied compounds was calculated using the following equation (Equation (1)):

$$Q_t = \frac{(C_o - C_t)V}{m} \tag{1}$$

where $Q_t$ (mg g$^{-1}$) is the adsorbed amount, i.e., the amount of carbofuran adsorbed per gram of the adsorbent; $C_o$ (mg L$^{-1}$), $C_t$ (mg L$^{-1}$), and V (L) are the initial concentration of adsorbate, the residual concentration at time t, and the total volume of the carbofuran, respectively; and m (g) is the mass of the adsorbent. Equation (2) is used to compute the carbofuran elimination efficiency (R).

$$\% \, R = \frac{(C_o - C_t)}{C_o} \, 100 \tag{2}$$

## 3. Results and Discussion

### 3.1. Metal Complex Inside the Nanopores of MIL-53-NH$_2$

The structural mechanism for the incorporation of metal ions in the MOF begins with the attack of the lone pair of the nitrogen atom in MIL-53-NH$_2$ on the carbonyl group in phthalic anhydride, resulting in the opening of the ring and release of the free carboxylic group (COOH) and amide group (NH-CO). The target molecule containing the free carboxylic group was treated with various metal ions (Fe(II), Cu(II) and Zn(II)) (Figure 1). MIL-53-NH-Ph has a modified chemical formula of Al(OH)(C$_8$H$_5$NO$_4$)$_{0.45}$ (C$_{16}$H$_9$NO$_7$)$_{0.55}$.

**Figure 1.** Post-synthetic modification of MIL-53-NH$_2$ with phthalic anhydride and its complexation with a transition metal.

### 3.2. Characterization

3.2.1. $^1$H NMR of MIL-53-NH-Ph

NMR spectroscopy was used to identify the number of protons, the chemical environment around different protons, and the formation of new compounds. The $^1$H NMR spectrum confirmed that MIL-53-NH$_2$ is linked to phthalic anhydride (Figure 2). The $^1$H NMR spectrum of MIL-53-NH-ph shows signals at = 7.01 (dd, 1H,), 7.58 (d, 1H,), 7.74 (d, 1H,) corresponded to the starting moiety, 2-amino terephthalic acid. The $^1$H NMR spectrum also displays multiplet signals at 7.65 (m, 2H), 7.94 (dd, 1H, 1H), and 8.03 (dd, 1H), corresponding to the phthalic anhydride ring (blue color in the spectrum). On the basis of the integration of the signals in the $^1$H NMR spectrum, the reaction ratio of MIL-53-NH$_2$ with phthalic anhydride was 54 percent.

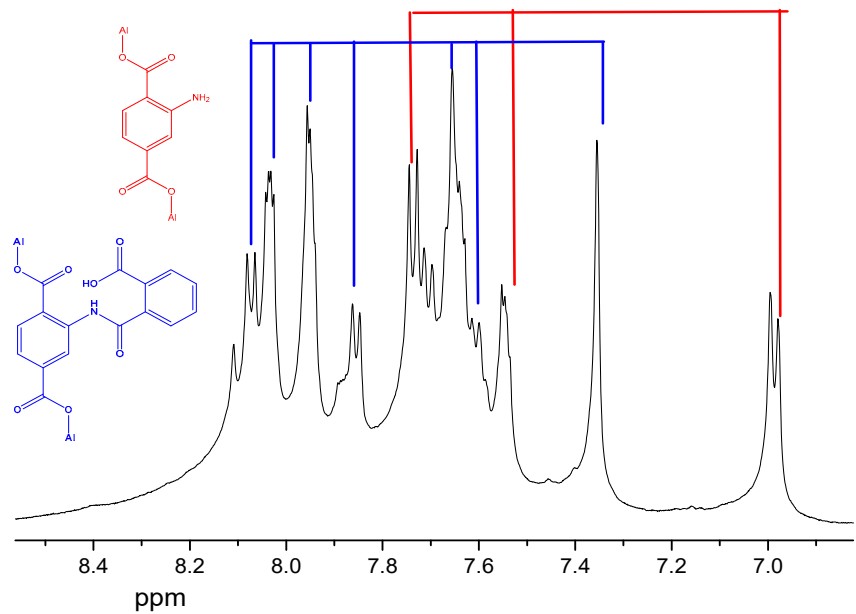

**Figure 2.** $^1$H NMR spectrum of MIL-53-NH-Ph in DCl + DMSO-d$_6$.

### 3.2.2. IR of the Synthesized Compounds

The FTIR spectrum of MIL-53-NH-Ph is similar to that of MIL-53-NH$_2$ except for a new peak at 1732 cm$^{-1}$ ascribed to the free carbonyl groups of the phthalic group arm (Figure 3). The FTIR data was used to confirm the subsequent synthetic alteration of MIL-53-NH$_2$ and to show the metal coordination domain (Figure 3). When MIL-53-NH-Ph is complexed, the C=O band at 1732 cm$^{-1}$ was shifted by 55 cm$^{-1}$ to a lower wavelength (1677 cm$^{-1}$), which is interpreted as evidence for C=O group coordination. Furthermore, new bands were displayed in the FTIR spectra of the complexes at 650 cm$^{-1}$ and 680 cm$^{-1}$, which were assigned to the M-O vibrations, indicating the formation of the complexes.

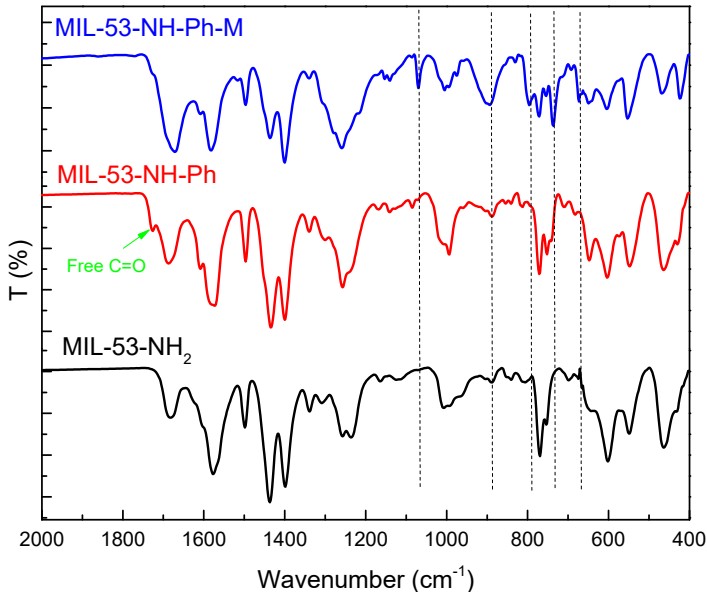

**Figure 3.** FTIR of MIL−53-NH$_2$, MIL−NH-Ph, and MIL−53-NH−Ph−M.

### 3.2.3. XRD and SEM analysis

The PXRD patterns of MIL-53-NH$_2$, MIL-53-NH-Ph, MIL-53-NH-Ph-Fe, MIL-53-NH-Ph-Zn, and MIL-53-NH-Ph-Cu are shown in Figure 4. All modified MIL-53-MOFs have PXRD patterns that are similar to that of MIL-53-NH$_2$. This indicates that the updated

MIL-53-MOFs were successfully prepared and showed the same 2θ peaks at 8.98°, 9.88°, 15.16°, 18.13°, 20.72, 25.23, 27.42, and 28.20°. This confirmed that the modified MOFs and pristine MOF have the same structure pattern without any structural modification.

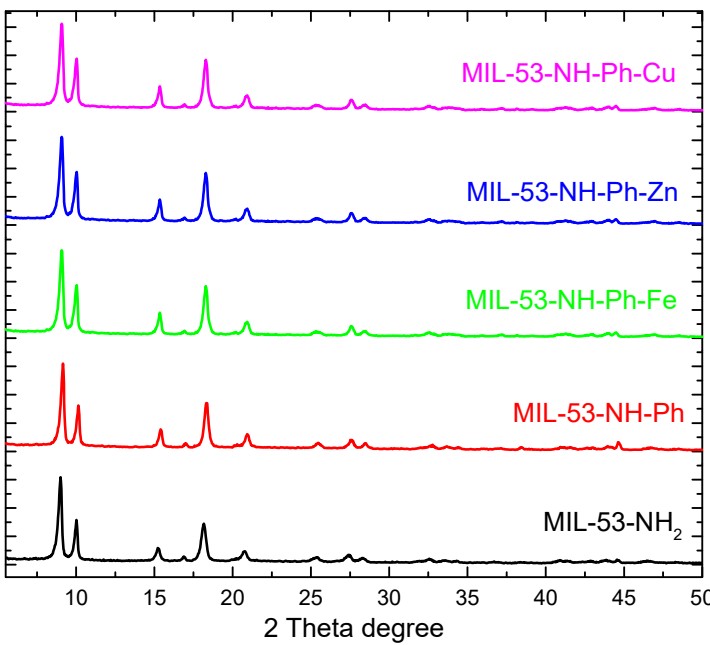

**Figure 4.** PXRD patterns of MIL−53−NH$_2$, MIL−53−NH−Ph, MIL−53−NH−Ph−Fe, MIL−53−NH−Ph−Zn, and MIL−53−NH−Ph−Cu.

Figure 5 displays SEM pictures of the modified MIL-53-MOFs, which demonstrate homogeneous morphology, and good dispersion with nanosize scale. The material's uniformity confirms its purity. The particle size of all the modified MIL-53-MOFs was around 50 nm. According to EDX analysis, the produced materials contain Al$^{3+}$, carbon, oxygen, and nitrogen. The modified materials have an iron ion peak in the EDX spectrum of MIL-53-NH-Ph-Fe, a copper ion peak in the EDX spectrum of MIL-53-NH-Ph-Cu, and a zinc ion peak in the EDX spectrum of MIL-53-NH-Ph-Zn. This indicates that the metal ions have been successfully incorporated into the MOF lattice.

The elemental analysis of MIl-53-NH$_2$, MIL-53-NH-ph, MIL-53-NH-ph-Fe, MIL-53-NH-ph-Zn, and MIL-53-NH-ph-Cu is given in Table 2. The carbon content was increased when MIl-53-NH$_2$ was modified with phthalic anhydride and then decreased with complexation with the metals. Interestingly, the ratio of aluminum ions decreases with the introduction of metal ions into MIL-53-NH-ph-Fe, MIL-53-NH-ph-Zn, and MIL-53-NH-ph-Cu. The surface areas of MIl-53-NH$_2$, MIL-53-NH-ph, MIL-53-NH-ph-Fe, MIL-53-NH-ph-Zn, and MIL-53-NH-ph-Cu were 1060, 950, 750, 725, and 730 m$^2$ g$^{-1}$, respectively. On the basis of these data, the surface area was decreased upon the complexation with metals.

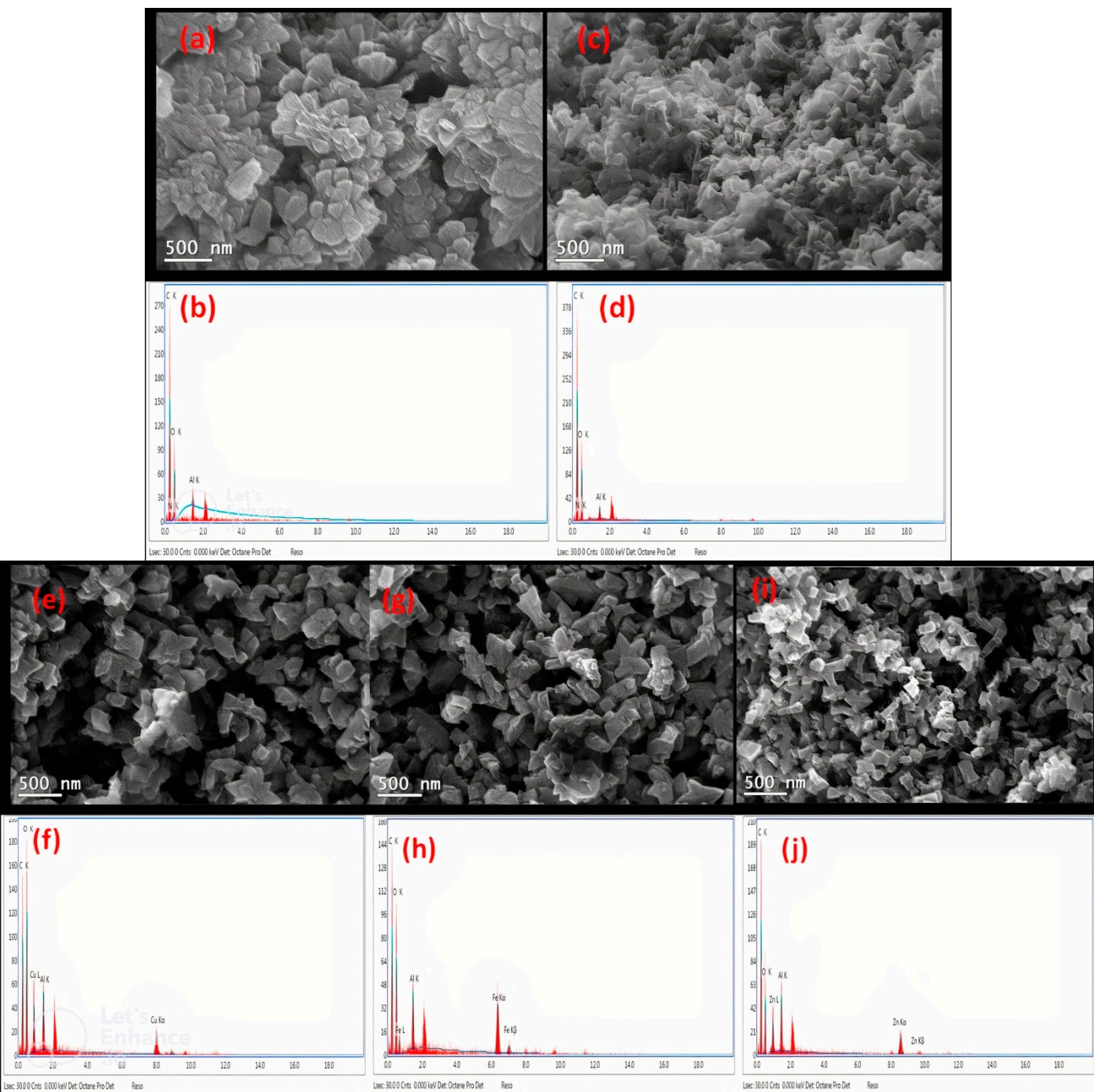

**Figure 5.** Scanning electron microscopic images and EDX spectra of (**a**,**b**) MIL-53-NH₂, (**c**,**d**) MIL-53-NH-Ph, (**e**,**f**) MIL-53-NH-Ph-Cu, (**g**,**h**) MIL-53-NH-Ph-Fe, and (**i**,**j**) MIL-53-NH-Ph-Cu.

**Table 2.** Elemental characteristics and surface properties of MIl-53-NH$_2$, MIL-53-NH-ph, MIL-53-NH-ph-Fe, MIL-53-NH-ph-Zn, MIL-53-NH-ph-Cu.

| Samples | Elemental Analysis (%) * | | | | | | | Surface Area (m$^2$ g$^{-1}$) | Pore Size (nm) | Pore Volume (cm$^3$ g$^{-1}$) |
|---|---|---|---|---|---|---|---|---|---|---|
| | C | H | N | Al | Zn | Fe | Cu | | | |
| MIl-53-NH$_2$ | 43.9 (43.07) | 2.55 (2.71) | 6.33 (6.28) | 12.3 (12.09) | - | - | - | 1060 | 0.8 | 0.5 |
| MIL-53-NH-ph | 48.51 (48.9) | 2.78 (2.71) | 4.75 (4.6) | 8.91 (8.86) | - | - | - | 950 | 0.7 | 0.47 |
| MIL-53-NH-ph-Fe | 44.98 (44.49) | 2.31 (2.30) | 4.25 (4.18) | 8.28 (8.06) | - | 9.5 (9.18) | - | 750 | 0.5 | 0.42 |
| MIL-53-NH-ph-Zn | 43.91 (43.81) | 2.14 (2.27) | 4.31 (4.12) | 8.1 (7.94) | 10.3 (10.5) | - | - | 725 | 0.44 | 0.41 |
| MIL-53-NH-ph-Cu | 44.1 (43.94) | 2.32 (2.27) | 4.12 (4.13) | 8.15 (7.96) | - | - | 10.5 (10.3) | 730 | 0.47 | 0.44 |

* The metal percentage was calculated from EDX. C, H, N percentages were measured for 3 samples, and the calculated value between brackets.

### 3.3. Adsorption Analysis Study

### 3.3.1. Effect of pH on the Adsorption Process

The effect of each synthesized material including MIL-53-NH$_2$ and all post-synthetic MOFs on the solution pH was intensively studied. The experiment was investigated at three pH values, 2, 7, and 9. In acidic solution (pH 2), each newly synthesized MOF and carbofuran were separately added to the acidic solution to show the effect of each material separately on the solution pH. It was found that the pH of the solution increased when adding MIL-53-NH$_2$, confirming the basic character of this compound, whereas, in the case of MIL-53-NH-ph, the pH decreased. The latter finding indicated that MIL-53-NH-ph has acidic characters, leading to an increase in the acidity of the solution. Furthermore, there was a slight lowering in the solution pH with the addition of MIL-53-NH-ph-Fe, MIL-53-NH-ph-Zn, MIL-53-NH-ph-Cu, and carbofuran, which asserted the acidic character of these compounds (Figure 6a). These compounds were also investigated in basic medium (pH = 9). The increased pH in the case of MIL-53-NH$_2$ was indicative of the high basicity of MIL-53-NH$_2$, while the pH declined in the case of MIL-53-NH-ph, which confirmed the results of the earlier former experiment. MIL-53-NH-ph-Fe, MIL-53-NH-ph-Zn, MIL-53-NH-ph-Cu, and carbofuran have acidic characters (Figure 6b). When the experiment was conducted at neutral pH (pH 7), no remarkable change in the pH of the solution was detected (Figure 6c). Therefore, pH 7 was suitable for achieving adsorption in this study. These materials are suitable for the treatment of drinking water due to maintaining the structure of MIL-53-MOFs and the presence of aluminum, iron, copper, and zinc ions was not detected in the solution after the adsorption experiment.

The effect of pH on adsorption was studied at various pH over the range of 3–11, and the solution pH was controlled using aqueous solutions of HCl (0.1 M) and NaOH (0.1 M). Figure 7 showed the influence of pH on carbofuran uptake using MIl-53-NH$_2$, MIL-53-NH-ph, MIL-53-NH-ph-Fe, MIL-53-NH-ph-Zn, and MIL-53-NH-ph-Cu. Significantly, the adsorption uptake of carbofuran was found to be pH-dependent, and the optimal pH was pH 7.

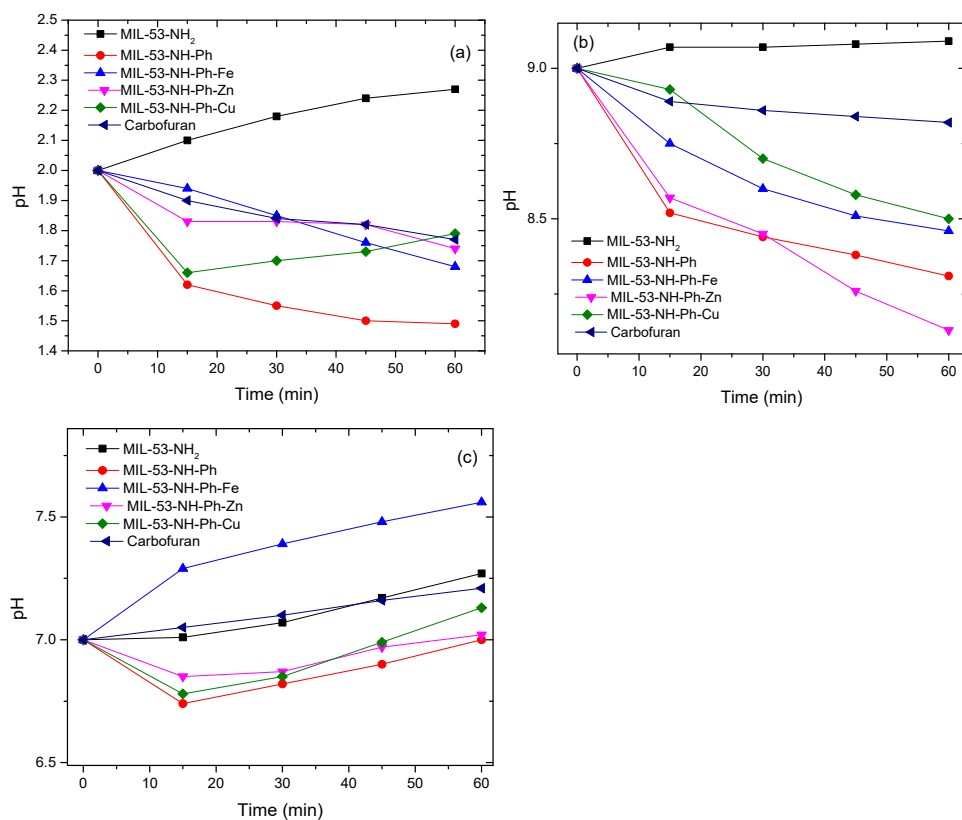

**Figure 6.** (**a**) Change in pH 2 profile (acidic medium) with time in the presence of MIl-53-NH$_2$, MIL-53-NH-ph, MIL-53-NH-ph-Fe, MIL-53-NH-ph-Zn, MIL-53-NH-ph-Cu, and carbofuran, (**b**) Change in pH 9 profile (basic medium) with time in the presence of MIl-53-NH$_2$, MIL-53-NH-ph, MIL-53-NH-ph-Fe, MIL-53-NH-ph-Zn, MIL-53-NH-ph-Cu, and carbofuran, (**c**) Change in pH 7 profile (neutral medium) with time in the presence of MIl-53-NH$_2$, MIL-53-NH-ph, MIL-53-NH-ph-Fe, MIL-53-NH-ph-Zn, MIL-53-NH-ph-Cu, and carbofuran.

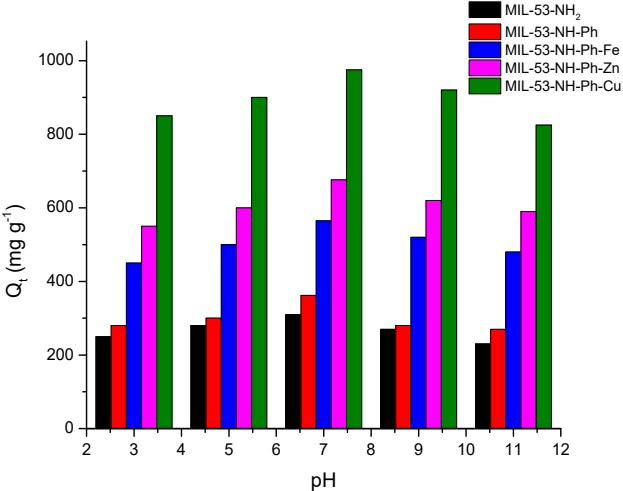

**Figure 7.** Effect of pH on the uptake of carbofuran onto MIl$-$53$-$NH$_2$, MIL$-$53$-$NH$-$ph, MIL$-$53$-$NH$-$ph$-$Fe, MIL$-$53$-$NH$-$ph$-$Zn, and MIL$-$53$-$NH$-$ph$-$Cu (Dosage: 30 mg L$^{-1}$, speed = 400 rpm, time: 120 min, pesticide concentration: 30 mg L$^{-1}$, and T = 303 k).

### 3.3.2. Effect of Pesticide Concentration

The concentration of carbofuran and the amount of carbofuran adsorbed on the MIl-53-NH$_2$, MIL-53-NH-ph, MIL-53-NH-ph-Fe, MIL-53-NH-ph-Zn, and MIL-53-NH-ph-Cu were

determined at room temperature using isotherm curves (Figure 8). The most commonly utilized adsorption models are the Freundlich and Langmuir models. The Langmuir model [16] can be expressed as a non-linear equation (Equation (3)):

$$Q_e = \frac{Q_m k_L C_e}{1 + k_L C_e} \tag{3}$$

where $C_e$ is the equilibrium concentration of carbofuran, $Q_e$ is the equilibrium adsorption capacity, $Q_m$ is the highest adsorption capacity, and $k_L$ is the Langmuir constant. The non-linear Freundlich model [17] is shown in Equation (4).

$$Q_e = k_F C_e^{\frac{1}{n}} \tag{4}$$

where $k_F$ is the Freundlich adsorption equilibrium constant and n is an empirical parameter.

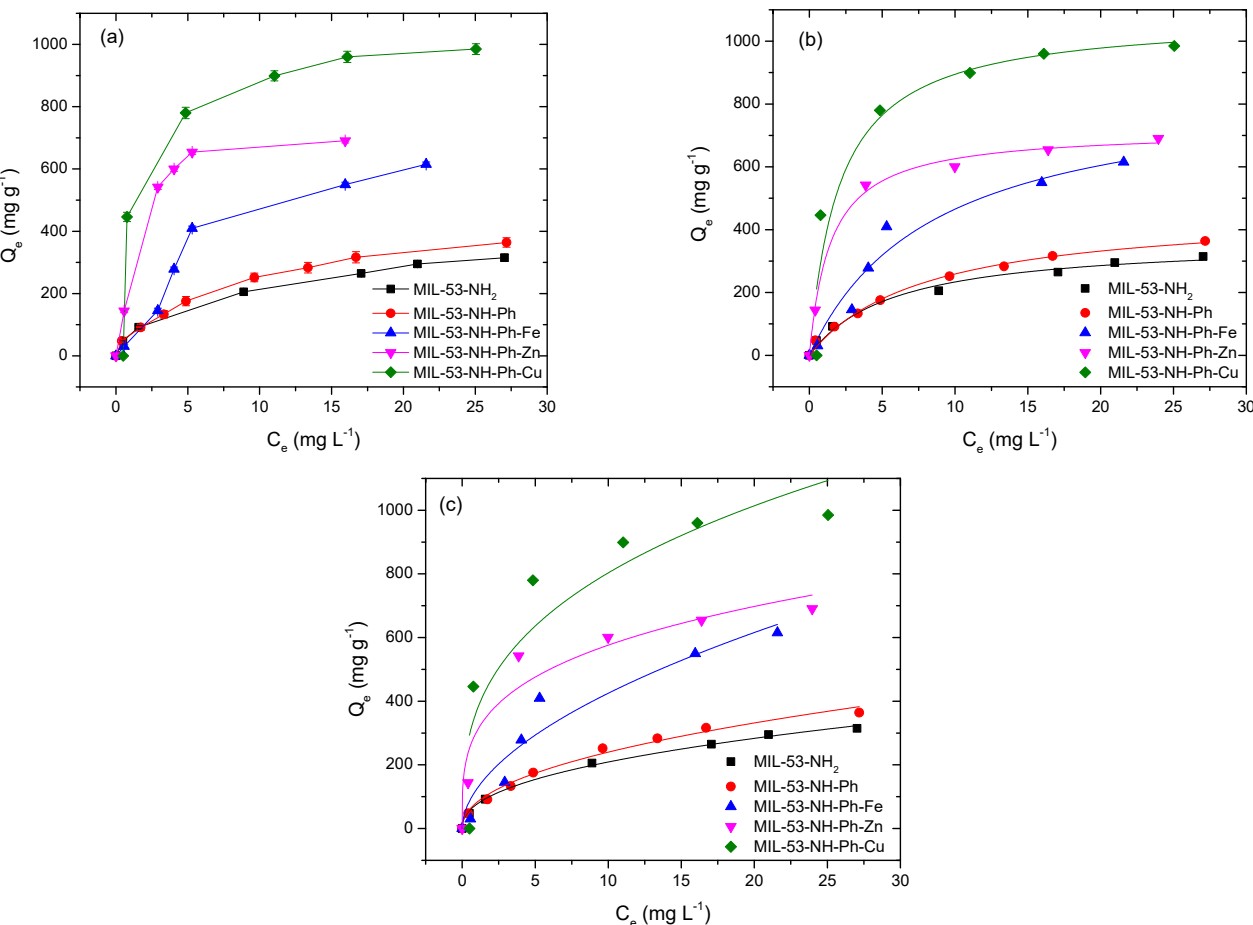

**Figure 8.** (**a**) Effect of carbofuran concentration on the uptake onto MIl$-$53$-$NH$_2$, MIL$-$53$-$NH$-$ph, MIL$-$53$-$NH$-$ph$-$Fe, MIL$-$53$-$NH$-$ph$-$Zn, and MIL$-$53$-$NH$-$ph$-$Cu; (**b**) Langmuir isotherm fitting for the modified MIL$-$53$-$MOFs; (**c**) Freundlich isotherm fitting for the modified MIL$-$53$-$MOFs. (Dosage: 30 mg L$^{-1}$, speed = 400 rpm, time: 120 min, pH: 7, and T = 303 k).

The Langmuir model proved to be appropriate for fitting the data based on the top correlation coefficient ($R^2$) (Table 3). This relates to the pesticide carbofuran adsorption on the adsorbent, which must therefore be monolayer adsorption.

**Table 3.** Adsorption isotherm parameters for the uptake of carbofuran onto MIL-53-NH$_2$, MIL-53-NH-ph, MIL-53-NH-ph-Fe, MIL-53-NH-ph-Zn, and MIL-53-NH-ph-Cu.

| Samples | Freundlich Parameters | | | | Langmuir Parameters | | | |
|---|---|---|---|---|---|---|---|---|
| | n | $k_F$ (mg g$^{-1}$) (mL mg$^{-1}$)$^{1/n}$ | $R^2$ | $\chi^2$ | $Q_m$ (mg g$^{-1}$) | $k_L$ (mL mg$^{-1}$) | $R^2$ | $\chi^2$ |
| MIL-53-NH$_2$ | 2.27 ± 0.09 | 76.08 ± 4.15 | 0.937 | 444.8 | 367.87 ± 4.58 | 0.173 ± 0.04 | 0.986 | 21.07 |
| MIL-53-NH-Ph | 2.14 ± 0.12 | 81.88 ± 5.96 | 0.949 | 167.7 | 462.06 ± 2.83 | 0.127 ± 0.014 | 0.992 | 18.04 |
| MIL-53-NH-Ph-Fe | 1.86 ± 0.33 | 124.00 ± 3.64 | 0.927 | 4322 | 662.94 ± 2.75 | 0.117 ± 0.037 | 0.958 | 85.19 |
| MIL-53-NH-Ph-Zn | 3.61 ± 0.86 | 304.92 ± 5.21 | 0.938 | 5216 | 717.62 ± 6.71 | 0.068 ± 0.09 | 0.995 | 36.03 |
| MIL-53-NH-Ph-Cu | 2.97 ± 0.98 | 370.27 ± 10.21 | 0.771 | 3427 | 978.64 ± 9.76 | 0.048 ± 0.24 | 0.988 | 14.34 |

### 3.3.3. Effect of Time on the Adsorption Process

The kinetics of carbofuran adsorption in MIL-53-NH$_2$, MIL-53-NH-Ph, MIL-53-NH-Ph-Fe, MIL-53-NH-Ph-Zn, and MIL-53-NH-Ph-Cu are shown in Figure 9. In both pseudo-first-order and pseudo-secondary kinetic models, the time-effect data were different at different time intervals (Table 4). The non-linear equation of the pseudo-first-order model [18] is written as follows:

$$Q_t = Q_e \left( 1 - \exp^{-k_1 t} \right) \tag{5}$$

The non-linear equation of the pseudo-second-order model is written as follows:

$$Q_t = \frac{k_2 Q_e^2 t}{1 + k_2 t Q_e} \tag{6}$$

where $Q_t$ is the adsorption capacity at time t, and $k_1$ and $k_2$ are the rates constant of the pseudo-first-order and pseudo-second-order kinetic models, respectively. A pseudo-second-order kinetic model ($R^2 = 0.997$) was found to be the best fit. The greatest carbofuran adsorption capabilities on MIL-53-NH$_2$, MIL-53-NH-Ph, MIL-53-NH-Ph-Fe, MIL-53-NH-Ph-Zn, and MIL-53-NH-Ph-Cu determined that MIL-53-NH-ph-Cu was the optimal material for carbofuran adsorption from wastewater.

**Table 4.** Kinetic parameters for carbofuran uptake onto MIl-53-NH$_2$, MIL-53-NH-ph, MIL-53-NH-ph-Fe, MIL-53-NH-ph-Zn, and MIL-53-NH-ph-Cu.

| Samples | $Q_e$ Exp. (mg g$^{-1}$) | Pseudo-First-Order Parameters | | | | Pseudo-Second-Order Parameters | | | |
|---|---|---|---|---|---|---|---|---|---|
| | | $Q_e$ (mg g$^{-1}$) | $k_1$ (min$^{-1}$) | $R^2$ | $\chi^2$ | $Q_e$ | $k_2 \times 10^{-5}$ (g mg$^{-1}$ min$^{-1}$) | $R^2$ | $\chi^2$ |
| MIL-53-NH$_2$ | 309 | 431.66 ± 12.92 | 0.0143 ± 0.001 | 0.926 | 370.6 | 315.42 ± 4.67 | 1.88 ± 0.02 | 0.997 | 28.99 |
| MIL-53-NH-Ph | 366 | 557.28 ± 26.83 | 0.0186 ± 0.003 | 0.938 | 270.1 | 369.11 ± 2.76 | 0.14 ± 0.02 | 0.992 | 49.04 |
| MIL-53-NH-Ph-Fe | 616 | 809.12 ± 32.21 | 0.0270 ± 0.001 | 0.946 | 145.3 | 619.76 ± 8.91 | 1.54 ± 0.17 | 0.997 | 10.91 |
| MIL-53-NH-Ph-Zn | 689 | 974.29 ± 34.54 | 0.0430 ± 0.002 | 0.966 | 201.1 | 673.56 ± 8.27 | 6.07 ± 1.16 | 0.986 | 16.12 |
| MIL-53-NH-Ph-Cu | 976 | 1041.4 ± 36.31 | 0.0492 ± 0.001 | 0.957 | 256.6 | 978.34 ± 4.89 | 1.88 ± 0.18 | 0.996 | 39.76 |

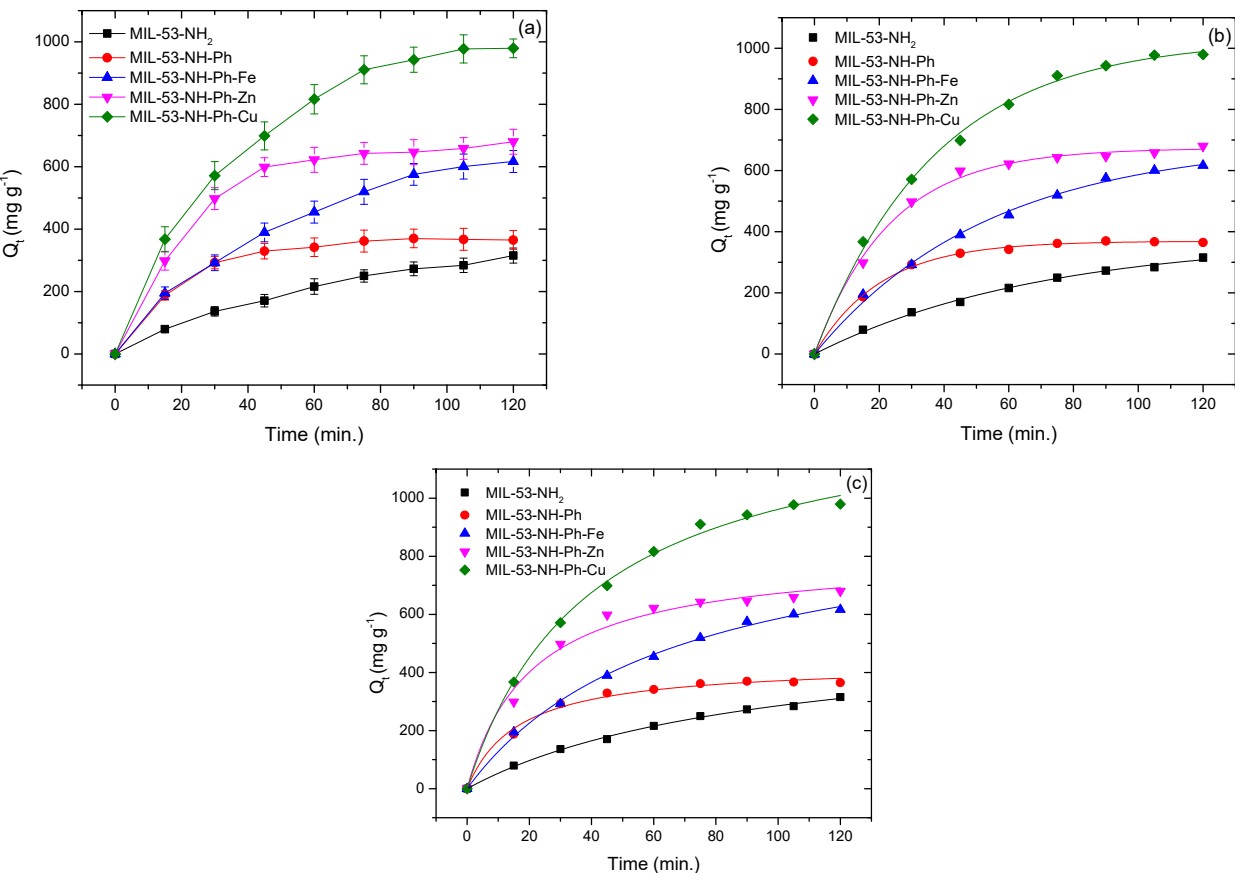

**Figure 9.** (**a**) Effect of time on adsorption of carbofuran onto MIL−53−NH₂, MIL−53−NH−ph, MIL−53−NH−ph−Fe, MIL−53−NH−ph−Zn, and MIL−53−NH−ph−Cu; (**b**) pseudo first order fitting; and (**c**) pseudo-second-order fitting. (Dosage: 30 mg L$^{-1}$, speed = 400 rpm, pesticide concentration: 30 mg L$^{-1}$, pH: 7, and T = 303 k).

*3.4. Adsorption Mechanism*

The π-π stacking interaction for the removal of carbofuran from wastewater is depicted in Figure 10. As the π-π system is present in both carbofuran insecticide and MIL-MOFs, the π-π stacking interaction is the dominant mechanism. The non-covalent interaction between carbon structures is studied using the π-π stacking process, which describes how two or more aromatic rings are linked. In molecular complexes containing two benzene rings, each forms a π-π stack and links with a benzene ring of carbofuran, and, thereby, intermolecular π-π stacking contact occurs. Each of the three oxygen atoms in carbofuran possesses one electron pair. The coordination bond is established when an electron pair is donated and accepted. Each oxygen creates a coordination bond with the metal, resulting in carbofuran and MOF structure coordination bonding. Furthermore, a hydrogen bond is established when a hydrogen atom attracts a more electronegative atom, such as an oxygen atom. In this system, the hydrogen of the amine group of MIL-MOFs and the oxygen of carbofuran form an intermolecular hydrogen bond.

**Figure 10.** The proposed adsorption mechanism of carbofuran onto the modified MIL-53-MOFs.

*3.5. Comparison of the Obtained Results with Previous Work*

Table 5 shows the maximum adsorption capabilities of various materials that adsorbed carbofuran. The composite materials proposed in this study had the highest adsorption when compared to other carbofuran-absorbable materials (See Table 5). It has been observed that MIL-53-NH-ph-Cu achieved an adsorption capacity three times that of activated carbon from rice straw and ten times that of commercial activated carbon. At the same time, the order of adsorption capacity of the materials in this study was MIL-53-NH-Ph-Cu > MIL-53-NH-Ph-Fe > MIL-53-NH-Ph-Zn > MIL-53-NH-Ph > MIL-53-NH$_2$. Therefore, MIL-53-NH-ph-Cu was the almost-perfect material for the adsorption of carbofuran from wastewater.

**Table 5.** Adsorption capacity for carbofuran onto different adsorbents reported in this study and the literature.

| Adsorbent | pH | $Q_m$ (mg g$^{-1}$) | Reference |
|---|---|---|---|
| MIL-53-NH-Ph-Cu | 7 | 978.6 | Current work |
| MIL-53-NH-Ph-Fe | 7 | 717.6 | Current work |
| MIL-53-NH-Ph-Zn | 7 | 662.9 | Current work |
| MIL-53-NH-Ph | 7 | 462.1 | Current work |
| MIL-53-NH$_2$ | 7 | 367.8 | Current work |
| Indian soils | 7 | 0.9–4.9 | [19] |
| Slow pyrolyzed sugarcane bagasse biochar | 6 | 3.6–18.9 | [20] |
| Animal bone meal | 6 | 18.5 | [21] |
| Tea waste biochars | 5 | 22.8–54.7 | [22] |
| Granular activated carbon | 7 | 96.2 | [23] |
| Commercial activated carbon | 7 | 97.1 | [24] |
| Date seed-activated carbon | 5.5 | 137.02 | [25] |
| Palm-oil-fronds-activated carbon | 7 | 164.0 | [26] |
| Magnetic sugarcane bagasse | 7 | 175.0 | [27] |
| Activated carbon from rice straw | 7 | 296.5 | [28] |
| Rice straw-derived activated carbon | 7 | 222.2–312.5 | [29] |
| Steel (blast furnace slag dust and sludge) | 7.5 | 208 | [30] |
| Orange peel | 7 | 84.49 | [31] |

## 4. Conclusions

MIL-53-NH$_2$, MIL-53-NH-ph, MIL-53-NH-ph-Fe, MIL-53-NH-ph-Zn, and MIL-53-NH-ph-Cu were all successfully synthesized. The herbicide carbofuran was removed from wastewater using the synthesized materials as effective adsorbents. MIL-53-NH$_2$ has a basic character, MIL-53-NH-ph has an acidic character, and MIL-53-NH-ph-Fe, MIL-53-NH-ph-Zn, and MIL-53-NH-ph-Cu have acidic characters. The proper kinetic model for carbofuran adsorption is pseudo-second order. The Langmuir isotherm model was shown to be a good fit for the equilibrium adsorption behavior. MIL-53-NH$_2$, MIL-53-NH-Ph, MIL-53-NH-Ph-Fe, MIL-53-NH-Ph-Zn, and MIL-53-NH-Ph-Cu had adsorption capacities of 367.8, 462.1, 662.94, 717.6, and 978.6 mg g$^{-1}$, respectively. $\pi$-$\pi$ stacking, coordination bonding, and hydrogen bonding were the adsorption mechanisms. The current study proposes an ideal approach for the removal of carbofuran from wastewater, on the basis that the framework has a free metal center that facilitates coordination bonding with carbofuran insecticide.

**Author Contributions:** M.N. worked on methodology, data curation, and writing—original draft, F.M.E., S.M.E.-M. and R.M.A.: Methodology, Data curation, Writing—Original draft preparation; Conceptualization, Supervision, Writing—Reviewing and Editing. All authors have read and agreed to the published version of the manuscript.

**Funding:** This research received no external funding.

**Data Availability Statement:** Not applicable.

**Conflicts of Interest:** The authors declare that they have no known competing financial interests or personal relationships that could have appeared to influence the work reported in this paper.

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
