# Peer review of "Remarkable Separation of Carbofuran Pesticide from Aqueous Solution Using Free Metal Ion Variation on Aluminum-Based Metal-Organic Framework"

_colloids, doi:10.3390/colloids6040073_

Round 1

Reviewer 1 Report

In general, the research content of this manuscript is novel, but it still needs minor revisions in order to make it better published. Comments and suggestions are listed below:

1. In the abstract section, the materials are named as 1, 2, 3, and 4, however this nomenclature is not used in the main text and in my opinion it is very helpful to use it instead of the complete names. Moreover, at the end of this part they express the units as mg g-1 and later on as mg/g, decide which one is better and unify it.

2. The introduction must be improved. At the begining the word pesticides is repited 4 consequtive times. In line 47, delete the brackets from (MOFs) and take of the word porous because it is also repeated. In this section is menctioned that MOFs materials are unsuitable for water treatment referencing the work nº 7 which is from 2011. In my opinion, it is not updated because there are many MOFs with improve properties for this final application (DOI: https://doi.org/10.1039/C6CS00362A). Line 58, start talking about ZIF-8 (Zn) but its reference and the corresponding to the following works are not identify or clear.

3. In the section, 2.2 it is not clear if all the reagents are dissolved separately.

4. The yield of the synthesized materials is not mentioned. And, how much amount of distilled water was used for the washings?

5. Unify the word FT-IR and FTIR, fig. or Figure.

6. With respect to reference 13, remove it or explain why it is here.

7. Justify why pH 7 is suitable to carry out the experiments.

8. In Figures 6, 7 and 8 are missing the letters a, b and c. In the case of 6 it is necessary also to define a, b and c in the figure caption.

9. Tables 1 and 2 show some parameters that are not defined and/or explained through equations, references nor definitions.

10. The authors use the words adsorption and absorption interchangeably, which imply very different processes or phenomena.

11. The Figure Caption of Figure 9 should start in capital letter.

12. The references of table 14 don’t correspond to the description of adsorbent which are referred to. Please, check all that apply.

 13. The conclusion has to be improved by emphasizing the results that reflect this work in an orderly way.

Author Response

Reviewer 1

In general, the research content of this manuscript is novel, but it still needs minor revisions in order to make it better published. Comments and suggestions are listed below:

Q1. In the abstract section, the materials are named as 1, 2, 3, and 4, however this nomenclature is not used in the main text and in my opinion it is very helpful to use it instead of the complete names. Moreover, at the end of this part they express the units as mg g-1 and later on as mg/g, decide which one is better and unify it.

Answer 1. Thank you for your recommendation, the abstract was corrected according to your comments

Q2. The introduction must be improved. At the begining the word pesticides is repited 4 consequtive times. In line 47, delete the brackets from (MOFs) and take of the word porous because it is also repeated. In this section is menctioned that MOFs materials are unsuitable for water treatment referencing the work nº 7 which is from 2011. In my opinion, it is not updated because there are many MOFs with improve properties for this final application (DOI: https://doi.org/10.1039/C6CS00362A). Line 58, start talking about ZIF-8 (Zn) but its reference and the corresponding to the following works are not identify or clear.

Answer 2. The introduction was updated, in terms of MOF stability, we agree with you that we use an old reference and we updated it according to your note, however, the stability of MOFs is very challenging and needs more work to reach the very stable material.

Q3. In the section, 2.2 it is not clear if all the reagents are dissolved separately.

Answer 3. This section was updated to be more understandable

Q4. The yield of the synthesized materials is not mentioned. And, how much amount of distilled water was used for the washings?

Answer 4. The yield of the product was added and more details about the synthetic procedures were added in the main text.

Q5. Unify the word FT-IR and FTIR, fig. or Figure.

Answer 5. It was corrected

Q6. With respect to reference 13, remove it or explain why it is here.

Answer 6. The references were updated

Q7. Justify why pH 7 is suitable to carry out the experiments.

Answer 7. Thank you for comment; we performed the effect of pH on the adsorption of carbofuran, at pH 7, the material has maximum adsorption uptake, all data and discussion was added in the main text.

Q8. In Figures 6, 7 and 8 are missing the letters a, b and c. In the case of 6 it is necessary also to define a, b and c in the figure caption.

Answer 8. The figures were updated with the key.

Q9. Tables 1 and 2 show some parameters that are not defined and/or explained through equations, references nor definitions.

Answer 9. The equations of isotherm and kinetic model were added in the main text with all identification for the parameters.

Q10. The authors use the words adsorption and absorption interchangeably, which imply very different processes or phenomena.

Answer 10. The text was updated

Q11. The Figure Caption of Figure 9 should start in capital letter.

Answer 11. It was corrected

Q12. The references of table 14 don’t correspond to the description of adsorbent which are referred to. Please, check all that apply.

Answer 12. The references in table 4 were checked and corrected

 Q 13. The conclusion has to be improved by emphasizing the results that reflect this work in an orderly way.

Answer 13. The conclusion was updated according to your note.

Reviewer 2 Report

Review for colloids-1906579

My review is centered around a scanned pdf of the manuscript that includes both linguistic suggestions and comments/question marks.

The detailed comments pertain to numbered items in that manuscript (#i) to which I will refer in order of appearance.

#1 … financially advantages systems for what? Context does not become clear.

#2 first I think the numbers in bold do not help here and I am not sure I understand what successfully means here. It could be omitted.

#3 hydrogen bond formation?

#4 the pH needs to be specified here. Moreover, since we deal with surface effects, normalization should be with respect to surface area. And for comparison with other substances it should be in molar units and not in mass.

#5 not clear what “commonly” means here. I think they are not commonly used in huge quantities, but worldwide there are huge quantities overall used. Please rephrase.

#6 not clear if the organ related toxicity only refers to humans. Please clarify and rephrase accordingly.

#7 incomplete sentence: challenging toxic substances?

#8 In these literature data, it is necessary to state the pH and compare on the basis moles per surface area. Mass per mass may be entirely misleading comparisons.

#9 why only one substrate was used in these sets of experiments.

#10 how was the pH controlled? Also I did not see a description of the pH-measurement set-up and calibration. Also it has to be stated whether carbon dioxide was excluded or not in the adsorption experiments.

#11  important solid properties like specific surface area and points of zero charge are missing. These are relevant characterization results in these kinds of studies. Moreover, I would expect a statement on the pH-dependent speciation of carbofuran.

#12 how was this all measured? ICP?

#13 no error bars in the figures and no indication of errors. This is required. Also in the caption at least the pH needs to be given. Again as pointed out above, the pH does not necessarily remain constant in these kinds of experiments as a function of added carbofuran and/or time. Without making sure the pH remains constant, the isotherms are no real isotherms, since pH has an effect on the uptake.

#14 units are missing for the parameters in the table. Also the numbers of digits sometimes are probably excessive. Please improve this table.

#15 again, the units are missing and also the equations for these models should be given (same for isotherm equations, where also the way of linearizing of the Langmuir isotherm needs to be specified). 

In all figures the pH needs to be specified. As pointed out, sufficient information is required for the read to understand the data and evaluate quality. Currently so much information is simply missing that the paper cannot be published.  

Author Response

Reviewer 2

My review is centered around a scanned pdf of the manuscript that includes both linguistic suggestions and comments/question marks. The detailed comments pertain to numbered items in that manuscript (#i) to which I will refer in order of appearance.

 Q1. financially advantages systems for what? Context does not become clear.

Answer 1. The text was modified according to your recommendation.

Q2 first I think the numbers in bold do not help here and I am not sure I understand what successfully means here. It could be omitted.

Answer 2. The text was updated according to your recommendation

Q3 hydrogen bond formation?

Answer 3. It was corrected

Q4 the pH needs to be specified here. Moreover, since we deal with surface effects, normalization should be with respect to surface area. And for comparison with other substances it should be in molar units and not in mass.

Answer 4. The effect of pH and surface area of prepared compounds were added in the text.

Q5 not clear what “commonly” means here. I think they are not commonly used in huge quantities, but worldwide there are huge quantities overall used. Please rephrase.

Answer 5. The sentence was modified

Q6 not clear if the organ related toxicity only refers to humans. Please clarify and rephrase accordingly.

Answer 6. The text was updated to be more understandable

Q7 incomplete sentence: challenging toxic substances?

Answer 7. The text was modified

Q8 In these literature data, it is necessary to state the pH and compare on the basis moles per surface area. Mass per mass may be entirely misleading comparisons.

Answer 8. The effect of pH on the adsorption uptake was performed and added in the main text.

Q9 why only one substrate was used in these sets of experiments.

Answer 9. Just for identification the modification of organic linker

Q10 how was the pH controlled? Also I did not see a description of the pH-measurement set-up and calibration. Also it has to be stated whether carbon dioxide was excluded or not in the adsorption experiments.

Answer 10. The pH controlled using HCl (0.1 N) and NaOH (0.1 N), and it was added in the main text.

Q11  important solid properties like specific surface area and points of zero charge are missing. These are relevant characterization results in these kinds of studies. Moreover, I would expect a statement on the pH-dependent speciation of carbofuran.

Answer 11. The surface area of prepared compounds was added in the main text. The pH-dependent speciation of carbofuran was added in the main text.

Q12 how was this all measured? ICP?

Answer 12. The detailed procedures were added in the part of the characterization, the measurements were analyzed using atomic absorption spectrophotometer.

Q13 no error bars in the figures and no indication of errors. This is required. Also in the caption at least the pH needs to be given. Again as pointed out above, the pH does not necessarily remain constant in these kinds of experiments as a function of added carbofuran and/or time. Without making sure the pH remains constant, the isotherms are no real isotherms, since pH has an effect on the uptake.

Answer 13. Thank you for your note, the error bars were added to the figures, the error bars are the SD of the data and the mean was drawn as a function of the model, the effect of pH on the adsorption uptake was added in the main text.

Q14 units are missing for the parameters in the table. Also the numbers of digits sometimes are probably excessive. Please improve this table.

Answer 14. The tables were modified according to your recommendation

Q15 again, the units are missing and also the equations for these models should be given (same for isotherm equations, where also the way of linearizing of the Langmuir isotherm needs to be specified). 

Answer 15. The equations of both isotherm and kinetic models were added and more identification for the parameters were also added in the main text.

Q16. In all figures the pH needs to be specified. As pointed out, sufficient information is required for the read to understand the data and evaluate quality. Currently so much information is simply missing that the paper cannot be published.  

Answer 16. Thank you very much for your effort in reading our manuscript carefully, all necessary data was added and we hope now it is suitable for publication.

Round 2

Reviewer 2 Report

I think this resubmission is very premature. The numbering of the article failed. The symbols for physical quantities should be homogenized so that they look the same in equations, explanations and in the tables or on figures. Out of respect for potential reviewers, the authors should take the time to re-read the version they (re)submit.

The rest of my review is centered around a scanned pdf of the manuscript that includes both linguistic suggestions and comments/question marks.

The detailed comments pertain to numbered items in that manuscript (#i) to which I will refer in order of appearance.

#1 the purpose is not mentioned

#2 this is confusing, experimental is meant to be uptake studies; spectroscopic studies are also experimental

#3 still not clear if this refers to humans or animals or both

#4 all these numbers are just numbers. for a comparison you would need much more detail, like the pH. You could put this in a table for example and use molar units so that the amount taken up can be compared. also it is necessary to compare with respect to surface area. I know I am asking for much, but we are not doing science just to repeat numbers on basis where a comparison is meaningless or to procude papers.

#5 N is big here and n is small in the equations. please take care to homogenize.

I think the authors shoud produce a mature manuscript before it should be judged. 

Author Response

Reviewer 2

Q1. I think this resubmission is very premature. The numbering of the article failed. The symbols for physical quantities should be homogenized so that they look the same in equations, explanations and in the tables or on figures. Out of respect for potential reviewers, the authors should take the time to re-read the version they (re)submit.

Answer 1. Thank you for your recommendation, we agree with you that the section numbering failed, but this is not our mistake, we submit the article in good format and we believe that the reviewers have no time to see this kind of error, this is not the first article we submit it for publication and we know that the reviewer time is much respect. For this article we submit the word file and the journal put it in the form of a journal and converted it to pdf and there is no chance for us to revise the pdf, this pdf was sent to you without our revision, I do not know the process of the journal, however, we provided the paper in good formatting as we usually submit it for the journals.

Q2. The rest of my review is centered around a scanned pdf of the manuscript that includes both linguistic suggestions and comments/question marks. The detailed comments pertain to numbered items in that manuscript (#i) to which I will refer in order of appearance.

Answer 2. Thank you very much for the time to print and correct and scan the documents with corrections, this is a great effort, usually overlooked by some judges. We worked on your notes, and we hope that the final version was satisfactory to your point of view.

Q3. The purpose is not mentioned

Answer 3. Thank you, the purpose was mentioned

Q4. This is confusing, experimental is meant to be uptake studies; spectroscopic studies are also experimental

Answer 4. Thank you, the sentence was updated

Q5. Still not clear if this refers to humans or animals or both

Answer 5. Thank you, the sentence was updated

Q6. All these numbers are just numbers. for a comparison you would need much more detail, like the pH. You could put this in a table for example and use molar units so that the amount taken up can be compared. also it is necessary to compare with respect to surface area. I know I am asking for much, but we are not doing science just to repeat numbers on basis where a comparison is meaningless or to procude papers.

Answer 5. Thank you, the table was added

Q7. N is big here and n is small in the equations. please take care to homogenize.

Answer 7. Thank you, the units and letters were updated
